# Relationship between Self-Reported Sleep Duration and Risk of Anemia: Data from the Korea National Health and Nutrition Examination Survey 2016–2017

**DOI:** 10.3390/ijerph18094721

**Published:** 2021-04-28

**Authors:** Min-Young Chun, Jeong-hoon Kim, Ju-Seop Kang

**Affiliations:** 1School of Biopharmaceutical and Medical Science, Sungshin Women’s University, Seoul 01133, Korea; sunrise91@sungshin.ac.kr; 2Department of Pharmacology & Clinical Pharmacology Lab, College of Medicine, Hanyang University, Seoul 04763, Korea; 3College of Medicine, American University of Antigua, University Park, Jabberwock Beach Road, Coolidge PO Box w 1451, Antigua and Barbuda; energeticjj@gmail.com

**Keywords:** anemia, sleep duration, premenopausal women, Korean population

## Abstract

The importance of sleep has been gaining more and more attention nowadays. It has been widely studied that some major health issues, such as cardiovascular diseases or mortality, are closely related to the extreme ends of sleep durations. Anemia is one of the health problems in modern society. In this study, we aimed to find a relationship between anemia occurrence and sleep duration. Data of 11,131 Korean adults aged 19 years or older were recruited from the 2016–2017 Korea National Health and Nutrition Examination Survey and analyzed in this cross-sectional study. ‘Anemia’ was defined in this study by hemoglobin level of <13 g/dL in men and <12 g/dL in women. Selected data were sorted into five groups by sleep duration: <5 h, 5 h ~ <6 h, 6 h ~ <8 h, 8 h ~ <9 h, and ≥9 h per day. We performed multivariate logistic regression analysis to assess the relationship between sleep duration and risk of anemia after adjusting for covariates including age, gender, family income level, education level, physical activity, cigarette smoking, and alcohol usage. Other factors were assessed in the analysis, such as depression, hypertension, diabetes, dyslipidemia, stroke, coronary artery disease, malignancy, stress level, and body mass index (BMI). We found that sleep duration of <5 h was related to high risk of anemia (odds ratio = 1.87; 95% confidence interval = 1.01–3.49, sleep duration of 6 h ~ <8 h as the reference group). Also, sleep duration of ≥9 h was related to lower risk of anemia in most premenopausal women after adjusting for covariates (odds ratio = 0.61; 95% confidence interval = 0.38–0.96, sleep duration of 6 h ~ < 8 h as the reference group). Male individuals with sleep durations of <5 h (odds ratio = 2.01; 95% confidence interval =1.05–3.84) and of ≥9 h (odds ratio = 2.48; 95% confidence interval =1.63–3.81) had a significantly higher risk of anemia without covariate adjustment. Postmenopausal women with sleep durations of ≥9 h had a significantly higher risk of anemia (odds ratio =2.02; 95% confidence interval =1.33–3.08) without adjusting for covariates. However, the associations became statistically insignificant after adjusting for age and covariates in both men and postmenopausal women. In conclusion, we found significant associations between extreme ends of sleep duration and risk of anemia in premenopausal Korean women. However, we did not observe strong associations between self-reported sleep duration and anemia risk in men or postmenopausal women.

## 1. Introduction

Anemia is highly prevalent in older individuals and has been linked to cardiovascular diseases, mortality, poor physical functionality, and disability [1,2,3]. Anemia is one of the well-known predictors of hospital mortality, especially among patients with cardiovascular diseases such as acute myocardial infarction, ischemic stroke, or congestive heart failure [4,5,6,7,8,9,10]. Anemia is also associated with reduced oxygen capacity, changes in blood viscosity, and impaired cerebral and cardiac regulation. [8,9,10] Given these negative impacts of anemia, identifying risk factors of anemia to formulate strategies to prevent and manage adverse outcomes quickly seems to be very beneficial. Of particular interest are factors suitable as intervention targets, with sleep duration one such modifiable risk factor.

Recently, many studies have reported that extreme sleep duration and sleep disturbance are independently associated with stroke, coronary heart disease, type 2 diabetes, and ultimately, increased mortality [11,12,13]. Some studies suggest that sleep duration is a causal risk factor of female infertility, gynecological disturbance, and distracted menstrual cycles caused by hormone augmentation or suppression in women [14,15]. Recently, few studies have examined the association between sleep and the risk of anemia. One evaluated the prevalence of anemia in the Chinese population and found that it was linked to sleep duration, with both short and long night sleep duration increasing the risk of anemia [16]. A British study reported women-specific associations between short sleep duration and risk of anemia [17].

However, even those extensive epidemiologic studies investigating the relationship between sleep duration and risk of anemia have certain limitations, particularly in adjusting for demographic factors and comorbidities. Therefore, in this study, we aimed to identify possible relationships between self-reported sleep duration and anemia in the Korean population. We further investigated potential relationships by narrowing down on the female population and dividing the sample according to their pre-menstrual or post-menstrual status. We used a national representative dataset drawn from the Korea National Health and Nutrition Examination Surveys (KNHANES) in 2016 and 2017. We hypothesized that sleep duration contributes significantly to the incidence of anemia and performed multivariate analyses to explore this relationship. Despite all the efforts, we could not clearly determine whether sleep duration is a causal risk factor for anemia based on this cross-sectional data. Therefore, further longitudinal studies are needed in the future.

## 2. Materials and Methods

### 2.1. Study Population

The data for our study were acquired from the seventh KNHANES (2016–2017). This survey is a nationally representative study of all ages and is designed to collect annual national data using stratified, multistage probability sampling to select household units. The KNHANES comprises a health interview survey, nutrition survey, a physical health examination survey, and clinical laboratory tests conducted by the Korea Center for Disease Control and Prevention (KCDC). [18] The health interview survey was conducted using self-reported structured questionnaires to obtain data, including sociodemographic information and economic characteristics, health status, service use, and behaviors. The survey information was gathered by visiting each household following agreement from the subjects, and physical health examinations were performed.

Among a total of 21,236 subjects (Figure 1), 16,277 completed the 2016–2017 KNHANES assessments, with an average response rate of 76.6%. Of the 16,277 people who participated in the 2016–2017 survey, 3377 individuals were aged <19 years old, and 1722 were missing information for variables analyzed in this study, leaving 11,178 adults aged ≥19 years old. We also excluded the 47 pregnant women to avoid possible bias caused by pregnancy influence on anemia. Thus, a total of 11,131 individuals were selected for the final analyses in this study. Additionally, we could not consider sex hormonal status or hormonal replacement therapy due to the lack of data available in this present study. Every KNHANES respondent provided written informed consent, and the Institutional Review Board of Sungshin Women’s University (Institutional Review Board No. SSWUIRB 2019–011) approved this study protocol.

### 2.2. Measurements

#### 2.2.1. Sleep-Duration Assessment

In this study, daily sleep time was obtained based on self-reported information in response to the question, “How many hours of sleep have you had on average?” The answer to this question was recorded as a numerical value (hours). Responses were categorized into five groups: <5 h, 5 h ~ <6 h, 6 h ~ <8 h, 8 h ~ <9 h, and ≥9 h per day, according to the mean sleep duration of 419 min (6.98 h) for the entire population.

#### 2.2.2. Anemia Assessment

Blood samples were taken from subjects who had fasted for more than 8 h, centrifuged, and transported to the Central Testing Institute, Seoul, Korea. After a maximum of 24 h in cold storage and analysis, hemoglobin was measured by the SLS hemoglobin detection method (XL-9000^®^; Sysmex, Hyogo, Japan). Anemia status (yes/no) was defined based on hemoglobin of less than 13 g/dL in men and less than 12 g/dL in women. KNHANES in 2016 and 2017 clarified the definition of anemia based on laboratory results; thus, we did not consider any clinical symptoms, nutritional status, and ongoing treatment.

#### 2.2.3. Sociodemographic and Health Behavior Variables

This study’s demographic and health behavior variables were age, sex, family income level, education, smoking status, alcohol use, and regular aerobic exercise. Demographic characteristics recorded by KNHANES included age, sex, and education level. Education level was stratified as none, elementary school, middle school, high school, and college or above. Household income was divided into four quartiles after adjusting for the number of families. Alcohol intake was classified as never, less than once per month, twice or four times per month, and over four times per month according to the frequency of drinking alcohol per month in the previous year. Smoking status was based on current cigarette smoking status, so that the data were divided into three classes: Lifelong nonsmoker, previous smoker, and current smoker. The clarification of the aerobic regular physical exercise (yes/no) is based on combining the types of exercise spread out throughout the week and their frequency. It is defined as moderate-intensity activity that the exercise with 64–76% of the maximum heart rate of at least 150 min per week. High-intensity exercise, more than 76% of the maximum heart rate, is required to perform at least 75 min per week, and it challenges communicating after a short period.

#### 2.2.4. Health Status Variables

The perceived stress level was assessed based on self-reported responses to the question “How much stress do you usually feel?” The responses were categorized as low (never or little), high, or very high. Depression was assessed based on the presence of diagnosed depression and answers to the question “Have you ever been diagnosed with depression by a doctor?” The response choices were binary (yes or no). Stroke, MI or angina pectoris, and cancer were also assessed based on physician diagnosis.

#### 2.2.5. Biological Variables

This study’s biological variables are based on the Korean National Nutritional Survey dataset released by KNHANES from Korea Disease Control and Prevention Agency in 2016–2017 by providing its guideline and clarifying its definition. Diabetes was defined based on a previous medical history of diabetes, a diagnostic tool for diabetes as fasting blood glucose of greater or equal to 126 mg/dL, or hemoglobin A1c of greater or equal to 6.5%, or the usage of prescription with glucose-lowering medication. Dyslipidemia was based on medical history, related medication use, total cholesterol (TC) of greater or equal to (≥) 240 mg/dL, triglycerides (TG) ≥ 200 mg/dL, LDL-C ≥ 160 mg/dL, or HDL-C ≤ 40 mg/dL. Fasting blood sugar (FBS), triglycerides (TG), high or low-density lipoprotein cholesterol (HDL-C, LDL-C), hs-CRP, and glycated hemoglobin (HbA1c) were measured using an automated chemistry analyzer (FBS, TG, HDL-C, LDL-C: Hitachi 7600–210, Hitachi, Japan; HbA1c: Tosoh G8, Tosoh, Japan; hs-CRP: COBAS, Roche, Germany). Hypertension was defined based on a previous history of hypertension, use of antihypertensive prescription, or the threshold of hypertension. It is described as patients with a blood pressure of greater or equal to 140 mmHg in systolic blood pressure, as well as greater or equal to 90 mmHg in diastolic blood pressure. Normal blood pressure has both SBP less than 120 mmHg and DBP of less than 80 mmHg. Trained examiners measured subjects’ height (cm) and weight (kg) to calculate body mass index (BMI). The standard value of BMI is greater or equal to 18.5 kg/m^2^ and less than 25 kg/m^2^. Either less than 18.5 kg/m^2^ or greater or equal to 25 kg/m^2^ is considered adjusted covariates.

### 2.3. Statistical Analysis

Subjects had been stratified into five groups (<5 h, 5 h ~ <6 h, 6 h ~ <8 h, 8 h ~ <9 h, and ≥9 h) based on the duration of their sleep period. Characteristics of the study population were analyzed using either a weighted chi-square test for categorical variables and a weighted one-way analysis of variance for continuous variables with Tukey or Bonferroni post hoc adjustment. A multivariable logistic regression analysis was executed to search for the association among the categorized groups after controlling for other covariates.

The included covariates were age, sex, family income, education, smoking, alcohol use, physical activity, BMI, stress levels, hypertension, diabetes, dyslipidemia, stroke, myocardial infarction or angina pectoris, and cancer. It has been a resource to determine a significant relationship with an individual’s sleep duration or anemia population. Model 1 is defined as unadjusted data. Model 2 is similar to model 1 except for adjusted age. Model 3 is the same as model 2 plus the adjusted demographics; different levels of education and family income, and health behavior factors included smoking, alcohol use, physical activity. Model 4 was adjusted for each demographic, health behavior factors, and health status factors had BMI, depression, stress level, hypertension, diabetes mellitus, dyslipidemia, stroke, myocardial infarction or angina pectoris, and cancer.

Five population groups in categories have been stratified as (<5 h, 5 h ~ <6 h, 6 h ~ <8 h (reference), 8 h ~ <9 h, and ≥9 h. It is included the adjusted odds ratios (ORs), and the corresponding 95% confidence intervals (CIs) upon each sleep durations and the presence of anemia. Furthermore, we performed the sub-analysis according to sex and menstrual status in women.

Using SPSS version 25 (SPSS Inc., Chicago, IL, USA), we analyzed population data statistically and clearly. In addition, two-sided *p*-values of less than 0.05 in data are considered statistically significant.

## 3. Results

The percentages of subjects who reported sleeping for <5 h, 5 h ~ <6 h, 6 h ~ <8 h, 8 h ~ <9 h, and ≥9 h per day were 4.4%, 10.1%, 56.7%, 19.5%, and 9.3%, respectively. Table 1 represents the characteristics of each baseline demographic stratified by sleep durations. An individual’s sleep duration is linked to each factor, including age, sex, family income, educational level, drinking status, smoking status, aerobic regular exercise, high stress, depression, stroke, diabetes mellitus, MI or angina pectoris, BMI, HbA1c, and hemoglobin.

Among the 11,131 subjects, 1001 (8.99%) had anemia (Table 2). The mean age of subjects with anemia was 51.55 ± 0.70 years old, and 20.0% were men. Participants with anemia were mostly older, more likely to be female, had lower family income, were less educated, reported higher stress (*p* = 0.015), and had higher HbA1c (*p* < 0.001), compared to those without anemia. Participants without anemia were more likely to be male, current smokers (*p* < 0.001), high-risk drinkers (*p* < 0.001), have higher physical activity (*p* < 0.001), higher BMI, higher cholesterol, and triglycerides, compared to those with anemia. Participants with anemia showed significantly higher prevalence of depression (*p* = 0.038), hypertension (*p* < 0.001), diabetes mellitus (*p* < 0.001), stroke (*p* = 0.018), MI or angina pectoris (*p* = 0.001), and cancer (*p* < 0.001).

Table 3 and Table 4 show multiple logistic regression models with statistical significance among sleep duration and anemia. Further analyses stratified by pre- or postmenstrual status among women are shown in Table 4. Compared with 6- <8-h sleep duration, self-reported sleep duration of greater or equal to 9 h was significantly associated with a higher risk of anemia in men (odds ratio = 2.48; 95% confidence interval = 1.63–3.81), without adjustments. However, the significance disappeared in the model adjusted for age and covariates. As shown in Table 4, subjects reporting a sleep duration of ≤5 h had a significantly higher risk of anemia (odds ratio = 1.87, 95% confidence interval = 1.01–3.49). Sleep duration ≥9 h was associated with lower risk of anemia, compared to greater than 6 h and less than 8 h sleeper in premenopausal women after adjusting for covariates (odds ratio 0.61, 95% confidence interval 0.38–0.96). Postmenopausal women showed a higher risk of anemia in the longest sleep duration group (≥9 h) in only unadjusted models (odds ratio = 2.02; 95% confidence interval = 1.33–3.08).

## 4. Discussion

In the present study, we observed associations between sleep duration and anemia in the Korean population, based on a dataset drawn from the KNHANES in 2016 and 2017. We found the associations between self-reported sleep duration and risk of anemia in premenopausal South Korean women. We found that the risk of anemia was higher in premenopausal women with extremely short sleep duration. On the other hand, the risk of anemia was lower in those with longer sleep duration than those who sleep 6 to 8 h. Significant associations were found between extreme short or long sleep duration and anemia in men and postmenopausal women only in unadjusted models. Still, they became clinically insignificant after adjusting for age and covariates. Therefore, we could not identify strong associations between self-reported sleep duration and risk of anemia in a Korean population.

As reported in Table 2, participants were more likely to have anemia if they were older, female, less educated, had lower family income, higher stress, higher HbA1c, and higher prevalence of depression, hypertension, diabetes mellitus, stroke, MI, or angina pectoris, and cancer, compared to those did not have anemia.

The Global Burden of Diseases (GBD 2010) study found that the prevalence of anemia decreased worldwide from 40.2% to 32.9% between 1990 and 2010 [19]. A previous study reported that the prevalence of anemia and moderate-severe anemias were 5.6% and 1.5%, respectively, in the US [20]. Since the prevalence of anemia has increased 4.0% to 7.1% and the prevalence of moderate to severe anemia has increased from 1.0% to 1.9%, nearly doubling from 2003 to 2012, anemia is considered a growing medical issue in the US [20]. The prevalence of anemia in the Korean population 10 years of age and over (age-standardized) decreased 2.7% from 9.5% in 2007 to 6.8% in 2018. Moreover, the prevalence in Korean women was 5.6-fold higher than that in men in 2018 [21]. So far, we do not have a general consensus regarding the exact estimate of the prevalence of anemia. However, anemia always has been a serious concern worldwide in the elderly population, who has been affected by adverse health concerns and hospital mortality. [1,2,3,4,5,6] Therefore, it would be beneficial to identify risk factors for anemia to determine effective intervention targets.

Our finding that women have a higher prevalence of anemia is consistent with previous studies [16,17,19,21], suggesting explanations such as menstrual bleeding, pregnancy, and gender differences in hormonal secretion, gastrointestinal absorption, nutrition, and psychosocial factors [16,20,22]. One review suggested sex difference-related hemoglobin levels are caused by the different effects of sex hormones on compensatory changes in kidney erythropoiesis, rather than direct effects on bone marrow [23].

Consistent with previous research regarding education level and household income, we observed that Koreans with higher educational levels and household incomes are less likely to experience anemia [24,25]. Higher education has been linked to higher household income, which helps individuals to pursue healthy lifestyles thus may lead to good health behaviors [26,27]. However, the relationship between socioeconomic status and health behaviors in Korea should be further explored. Previous research reported that poor psychological status is more prevalent in anemia and suggested that psychological problems are essential in the development of anemia [21,27]. Aligning with the results of this study, other studies previously reported that stress and depression significantly affect the outcome of anemia and low hemoglobin level [21,28].

Extreme long sleep duration increases inflammation and elevates hs-CRP level. [17,29,30,31]. In the Whitehall II study, CRP was elevated in women with extreme short sleep duration after adjusting for covariates, while no association was found in men. [32] However, our study failed to find any associations between sleep duration and CRP levels as an inflammatory marker. [33] 

Previous studies that had investigated the relationship between sleep and anemia have reported varying results [16,17]. This variation is likely to originate from differences in methodology, including selecting target subjects, measurement methods, and adjustments for covariates. Previous studies have been limited by small sample sizes and confounding demographic factors [16,17]. In the present study, we identified a relationship between sleep duration and anemia in a large Korean population-based sample. By introducing a few confounding factors such as menopausal status or long-standing illness related to anemia and sleep, we could increase the power of our findings. In an English longitudinal study [17], which targeted a community-dwelling population aged 50 and older, the extreme ends of sleep duration and sleep problems were associated with lower hemoglobin levels in men, independent of covariates. More disturbed sleep was related to a higher risk of anemia in both sexes [17]. However, no association was found between anemia status and differences in sleep durations in the British older population [17]. In a Chinese study, both long and short sleep duration were independent predictors of incident anemia after a follow-up of 8 years [16].

In the present study, we did not find a robust association between self-reported sleep duration and risk of anemia in both sexes (Table 3). Despite this fact, in our further analyses stratified by pre- or post-menstrual status among women, we found significant differences in the relationships between the subgroups and anemia. Pre-menopausal women who sleep less than or equal to 5 h were more likely to have anemia, while those who sleep 9 h, or more were less likely to have anemia. 

Because we used cross-sectional data, it remains unclear whether short sleep duration acts as a risk factor or is a consequence of lower hemoglobin levels and anemia. Therefore, further studies are needed to compare sleep duration before and after anemia treatment. A few studies have examined the possible mechanisms underlying the relationship between sleep duration and anemia status [16,17]. Small experimental studies indicated that sleeping raises androgen levels and causes a direct stimulatory effect in the bone marrow associated with erythropoietin and erythropoietin production in the kidney [34,35]. We did not observe a robust association between short sleep duration and risk of anemia in males. However, there have been a few studies reporting that short sleep hours are associated with problems such as sleep apnea and snoring, which, in turn, are related to the risk of anemia in men [17]. These observations suggest that men are more to sleep disorders of poor sleep quality than sleep duration for anatomical and physical reasons.

The NIH State of the Science Panel’s Conference Statement indicates that menopausal women seem to have more sleep difficulty as they progress through menopausal stages [36,37]. The prevalence of sleep disturbance varies from 16 to 42% in premenopausal women, from 39 to 47% in perimenopausal women, and from 35 to 60% in postmenopausal women. These sleep disturbances can cause considerable stress, reduce the quality of life, increase healthcare costs, induce disability, and exacerbate medical and psychiatric conditions [38]. Previous studies reported that women are more likely to experience sleep disturbance than men due to the increase in IL-6 and CRP levels, which serve as inflammatory factors in the body. [36,37,38] Thus, it is crucial to understand the inflammatory effects of sleep disturbance in women [39]. Women are more likely to experience periods of poor sleep, especially during midlife and menopause, due to hot flashes and mood disorders. Therefore, extremely long sleep duration may be related to poor sleep continuity, such as fragmented sleep, which might increase the risk of anemia among postmenopausal women [37,38].

The present study has several limitations. First, because we used data obtained from KNHANES, a cross-sectional survey, this study design did not establish temporal precedence between many factors and anemia. Second, since sleep duration was assessed using self-reported measures, it is prone to subjective bias. However, previous research based on a large community population established that there is generally a favorable agreement between objective measurements and subjectively assessed sleep [40,41]. Third, because anemia was diagnosed solely based on hemoglobin content in this study, we could not consider different types of anemia, nutritional status, and ongoing treatment. Finally, the possibility of unmeasured variables, incomplete adjustment of covariates, or mediating effect of other covariates on the relationship between sleep duration and incidence of anemia cannot be excluded. Therefore, further study is needed to analyze the mediation effect of various covariates through structural equation modeling. Despite these limitations, our results could be generalized to a large population because our findings were based on a nationally representative sample.

In conclusion, we found the presence of an association between self-reported sleep duration and risk of anemia in premenopausal South Korean women. However, we did not observe robust associations between self-reported sleep duration and risk of anemia in men or in postmenopausal women. Thus, additional studies should focus on postmenopausal women. Healthcare professionals who treat patients with anemia, especially premenopausal women, in routine clinical practice should be aware of the associations between anemia and sleep problems. Specific assessment and treatment strategies addressing sleep problems should be included in anemia treatment.

## Figures and Tables

**Figure 1 ijerph-18-04721-f001:**
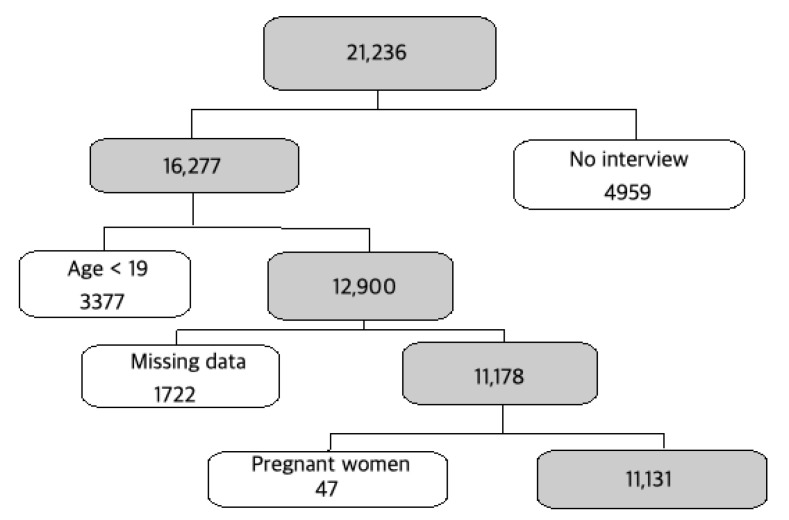
Flowchart showing inclusion and exclusion of subjects according to study criteria.

**Table 1 ijerph-18-04721-t001:** Demographics of the study participants according to sleep duration.

Variable	Sleep Duration (hours)					*p*-Value *
	<5 h	5 h ~ <6 h	6 h ~ <8 h	8 h ~ <9 h	≥9 h	
Questionnaire-Based Data	(*n* = 492)	(*n* = 1129)	(*n* = 6147)	(*n* = 2226)	(*n* = 1137)	
Age, years	47.12 ± 1.04 ^ab^	47.76 ± 0.59 ^ab^	46.28 ± 0.3 ^a^	46.88 ± 0.48 ^a^	48.88 ± 0.82 ^bc^	0.002
male, sex, *n* (%)	206 (47.7) ^ab^	461 (46.5) ^a^	2821(52.7) ^b^	950(47.4) ^ab^	476(45.7) ^a^	<0.001
Family income, *n* (%) ,						
Low	134 (23.5)	220 (16.0)	940 (12.5)	435 (16.1)	380 (28.4)	<0.001
Lower middle	123 (24.6)	269 (23.0)	1433 (22.1)	579 (24.5)	284 (23.5)	
Upper middle	135 (30.4)	308 (28.9)	1760 (30.8)	602 (29.0)	268 (26.4)	
High	99 (21.5)	331 (32.1)	1997 (34.6)	604 (30.4)	201 (21.6)	
High school or above , *n* (%)	316 (72.8) ^ab^	804 (78.0) ^b^	4603 (81.3) ^b^	1514 (76.1) ^ab^	630 (66.3) ^a^	<0.001
Current smoker , *n* (%)	103 (24.0) ^a^	217 (21.4) ^a^	1158 (22.8) ^b^	363 (19.6) ^a^	197 (21.6) ^a^	0.043
Frequency of alcohol intake/month, * n * (%)						
Never	103 (21.7)	182 (15.4)	895 (13.4)	336 (14.4)	221 (18.6)	0.021
Less than once	108 (27.9)	314 (31.4)	1768 (30.9)	637 (31.9)	299 (30.6)	
Two– four times	86 (21.0)	260 (26.8)	1456 (28.8)	497 (28.0)	204 (23.4)	
Over four times	115 (29.4)	248 (26.4)	1417 (26.9)	472 (25.7)	244 (27.3)	
Aerobic regular exercise, *n* (%)	211 (45.9) ^ab^	512 (49.0) ^a^	2849 (49.0) ^a^	941 (46.4) ^ab^	388 (39.5) ^b^	<0.001
High stress or above , *n* (%)	47 (10.5) ^a^	56 (5.0) ^bc^	273 (4.3) ^c^	93 (4.6) ^b^	54 (5.4) ^b^	<0.001
Depression, *n* (%)	37 (7.5) ^a^	50 (4.0) ^ab^	241 (3.5) ^b^	95 (3.9) ^b^	63 (5.1) ^ab^	<0.001
Hypertension, *n* (%)	143 (24.5) ^a^	284 (20.1) ^ab^	1325 (17.4) ^b^	576 (19.6) ^ab^	352 (24.7) ^a^	<0.001
Diabetes mellitus, *n* (%)	62 (10.0) ^a^	99 (7.5) ^ab^	551 (7.2) ^b^	223 (7.8) ^ab^	148 (9.8) ^a^	0.026
Dyslipidemia, *n* (%)	107 (20.2)	215 (15.5)	1059 (14.9)	420 (15.7)	198(14.7)	0.071
Stroke, *n* (%)	7 (0.8) ^a^	18 (1.5) ^a^	113 (1.4) ^a^	63 (2.0) ^a^	44 (3.4) ^b^	<0.001
MI or Angina, *n* (%)	12 (1.4) ^ab^	19 (1.1) ^a^	105 (1.3) ^ab^	55 (2.0) ^b^	31 (2.3) ^b^	0.010
Cancer, *n* (%)	12 (1.7)	17 (1.3)	128 (1.7)	47 (1.8)	30 (2.1)	0.667
Exam-Based Data						
BMI (kg/m^2^)	24.70 ± 0.22 ^a^	24.28 ± 0.11 ^b^	23.95 ± 0.06 ^b^	23.80 ± 0.10 ^bc^	23.58 ± 0.13 ^c^	<0.001
HbA1c (%)	5.80 ± 0.06 ^a^	5.69 ± 0.03 ^b^	5.61 ± 0.01 ^b^	5.63 ± 0.02 ^b^	5.66 ± 0.03 ^ab^	0.002
Total cholesterol (mg/dL)	197.36 ± 2.16	194.71 ± 1.15	193.29 ± 0.60	192.80 ± 0.88	191.77 ± 1.36	0.159
HDL-cholesterol	51.67 ± 0.65	51.35 ± 0.48	51.26 ± 0.21	50.92 ± 0.29	50.65 ± 0.43	0.522
Triglycerides	154.23 ± 10.72	139.81 ± 4.45	138.73 ± 2.86	138.73 ± 2.86	138.91 ± 3.31	0.708
LDL-cholesterol	121.59 ± 4.53	121.52 ± 3.02	118.59 ± 1.35	118.50 ± 2.36	120.88 ± 3.20	0.837
Hemoglobin (g/dL)	14.16 ± 0.10 ^a^	14.12 ± 0.06 ^a^	14.29 ± 0.03 ^b^	14.16 ± 0.04 ^a^	14.06 ± 0.06 ^a^	<0.001
hs-CRP (mg/L)	1.27 ± 0.10	1.16 ± 0.06	1.14 ± 0.03	1.21 ± 0.05	1.29 ± 0.07	0.162

Values are expressed as mean ± standard error for continuous variables and numbers (%) for categorical variables. *p*-values * are determined by weighted chi-square tests of categorical variables and by weighted analysis of variance of continuous variables between all groups. BMI: body mass index, HbA1c: hemoglobin A1c, HDL: high-density lipoprotein, LDL: low-density lipoprotein. hs-CRP: high-sensitivity C-reactive protein. Different superscript indicates significant (*p* < 0.05) post hoc pairwise difference using Tukey (for continuous variable) or Bonferroni (for categorical variable) correction.

**Table 2 ijerph-18-04721-t002:** Demographics of the study participants with anemia and without anemia.

Variables	Anemia (*n* = 1001)	Without Anemia (*n* = 10,130)	*p*-Value *
Questionnaire-Based Data			
Age, years	51.55 ± 0.7	46.48 ± 0.23	<0.001
male, sex, *n* (%)	234 (20.0)	4680 (52.7)	<0.001
Family income level , *n* (%)			<0.001
Low	268 (22.2)	1841 (15.0)	
Lower middle	261 (25.8)	2427 (22.7)	
Upper middle	239 (25.4)	2834 (30.2)	
High	232 (26.6)	3000 (32.1)	
High school or above , *n* (%)	638 (70.6)	7229 (78.8)	<0.001
Current smoker , *n* (%)	65 (6.8)	1973 (23.3)	<0.001
Frequency of alcohol intake per month, * n * (%)			<0.001
Never	227 (25.6)	1510 (13.8)	
Less than once	288 (37.4)	2838 (30.5)	
Two– four times	184 (23.5)	2319 (27.9)	
Over four times	122 (13.5)	2374 (27.8)	
Aerobic regular exercise, *n* (%)	380(41.2)	4521(48.0)	0.001
High stress or above , *n* (%)	53 (5.8)	440 (5.0)	0.015
Depression, *n* (%)	56 (5.2)	430 (3.9)	0.038
Hypertension, *n* (%)	308 (25.8)	2405 (18.9)	<0.001
Diabetes mellitus, *n* (%)	175 (15.1)	905 (7.1)	<0.001
Dyslipidemia, *n* (%)	180(16.8)	1819 (15.2)	0.218
Stroke, *n* (%)	33 (2.7)	212 (1.6)	0.018
MI or Angina pectoris, *n* (%)	47 (3.6)	273(2.0)	0.001
Cancer , *n* (%)	48 (4.2)	186 (1.5)	<0.001
Exam-Based Data			
BMI (kg/m^2^)	22.85 ± 0.12	24.04 ± 0.05	<0.001
HbA1c (%)	5.76 ± 0.03	5.62 ± 0.01	<0.001
Total cholesterol (mg/dL)	181.56 ± 1.34	194.38 ± 0.49	<0.001
HDL-cholesterol (mg/dL)	51.91 ± 0.51	51.10 ± 0.16	0.119
Triglycerides (mg/dL)	102.61 ± 2.33	142.90 ± 1.88	<0.001
LDL-cholesterol (mg/dL)	110.92 ± 4.41	119.53 ± 1.04	0.059
Hemoglobin (g/dL)	11.14 ± 0.04	14.48 ± 0.02	<0.001
hs-CRP (mg/L)	1.29 ± 0.08	1.17 ± 0.02	0.139

Values are expressed as mean ± standard error for continuous variables and numbers (%) for categorical variables. *p*-values * are determined by weighted chi-square tests of categorical variables and by weighted *t*-test of variance of continuous variables between two groups. BMI: body mass index, HbA1c: hemoglobin A1c, HDL: high-density lipoprotein, LDL: low-density lipoprotein. hs-CRP: high-sensitivity C-reactive protein.

**Table 3 ijerph-18-04721-t003:** Crude and adjusted odds ratios for sleep duration associated with anemia according to various statistical models.

Sleep Duration (Hours)					
Total	<5 h	5 h ~ <6 h	6 h ~ <8 h	8 h ~ <9 h	≥9 h
Model 1 ^a^	1.37 (0.96–1.96)	1.25 (0.75–1.59)	reference	1.04 (0.85–1.28)	1.29 (1.00–1.65)
Model 2 ^b^	1.25 (0.87–1.43)	1.11 (0.86–1.43)	reference	0.97 (0.79–1.19)	1.14 (0.89–1.46)
Model 3 ^c^	1.12 (0.94–2.13)	1.08 (0.82–1.43)	reference	0.95 (0.75–1.19)	1.07 (0.80–1.43)
Model 4 ^d^	1.60 (1.05–2.39)	1.14 (0.86–1.51)	reference	0.91 (0.72–1.15)	1.01 (0.76–1.36)
Women					
Model 1 ^a^	1.14 (0.75–1.74)	1.13 (0.86–1.48)	reference	0.85 (0.68–1.07)	0.94 (0.69–1.28)
Model 2 ^b^	1.14 (0.75–1.74)	1.12 (0.85–1.47)	reference	0.85 (0.68–1.01)	0.93 (0.69–1.27)
Model 3 ^c^	1.37 (0.85–2.19)	1.08 (0.80–1.470	reference	0.82 (0.64–1.06)	0.85 (0.60–1.20)
Model 4 ^d^	1.51 (0.93–2.44)	1.10 (0.81–1.49)	reference	0.81(0.63–1.05)	0.84 (0.60–1.19)
Men					
Model 1 ^a^	2.01 (1.05–3.84)	1.23 (0.67–2.25)	reference	1.57 (1.07–2.31)	2.48 (1.63–3.81)
Model 2 ^b^	1.75 (0.90–3.37)	1.23 (0.66–2.28)	reference	1.22 (0.81–1.85)	1.34 (0.84–2.11)
Model 3 ^c^	1.52 (0.72–3.20)	1.29 (0.68–2.47)	reference	1.14 (0.72–1.82)	1.15 (0.58–1.93)
Model 4 ^d^	1.51 (0.72–3.15)	1.49(0.78–2.87)	reference	1.16 (0.72–1.89)	1.06 (0.62–1.79)

Data are presented as estimated marginal means (95% confidence intervals). Model 1 ^a^: None of the variables are adjusted. Model 2 ^b^: Age is adjusted. Model 3 ^c^: Age, level of education, family income level, smoking, alcohol, and physical activity are adjusted. Model 4 ^d^: Model 3 plus stress, depression, hypertension, diabetes mellitus, dyslipidemia, stroke, myocardial infarction or angina pectoris, and body mass index are adjusted.

**Table 4 ijerph-18-04721-t004:** Crude and adjusted odds ratios for sleep duration associated with anemia stratified menstrual status in women.

Total	Sleep Duration (Hours)				
	<5 h	5 h ~ <6 h	6 h ~ <8 h	8 h ~ <9 h	≥9 h
Premenopausal State					
Model 1 ^a^	1.31 (0.72–2.38)	1.32 (0.92–1.88)	reference	0.89 (0.68–1.19)	0.53 (0.35–0.83)
Model 2 ^b^	1.34 (0.72–2.47)	1.27 (0.89–1.83)	reference	0.92 (0.69–1.21)	0.59 (0.38–0.92)
Model 3 ^c^	1.62 (0.87–3.05)	1.28 (0.88–1.85)	reference	0.86 (0.64–1.16)	0.61 (0.38–0.96)
Model 4 ^d^	1.87 (1.01–3.49)	1.31(0.89–1.91)	reference	0.85 (0.63–1.45)	0.61 (0.38–0.96)
Postmenopausal State					
Model 1 ^a^	1.13 (0.69–1.91)	0.96 (0.59–1.57)	reference	0.82 (0.55–1.24)	2.02 (1.33–3.08)
Model 2 ^b^	0.96 (0.55–1.68)	0.93 (0.56–1.52)	reference	0.76 (0.51–1.15)	1.51 (0.99–2.43)
Model 3 ^c^	1.12 (0.54–2.31)	0.76 (0.42–1.39)	reference	0.71 (0.41–1.21)	1.49 (0.82–2.71)
Model 4 ^d^	1.10 (0.53–2.32)	0.80 (0.43–1.49)	reference	0.69 (0.40–1.19)	1.51 (0.82–2.79)

Data are presented as estimated marginal means (95% confidence intervals). Model 1 ^a^: None of the variables are adjusted. Model 2 ^b^: Age is adjusted. Model 3 ^c^: Age, level of education, family income level, smoking, alcohol, and physical activity are adjusted. Model 4 ^d^: Model 3 plus stress, depression, hypertension, diabetes mellitus, dyslipidemia, stroke, myocardial infarction or angina pectoris, and body mass index are adjusted.

## Data Availability

Data set presented in this study is available on request to Korea Center for Disease Control and Prevention (KCDC).

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
