# Peer review of "Relationship between Self-Reported Sleep Duration and Risk of Anemia: Data from the Korea National Health and Nutrition Examination Survey 2016–2017"

_ijerph, 2021, doi:10.3390/ijerph18094721_

Round 1
Reviewer 1 Report
Please find an attached review.

Author Response
"I would like to extend my deep appreciation to you for review. I thankfully found this round of editing a good chance of improving the paper."

Reviewer 2 Report
The authors have made lots of improvement comparing to the original version. However, the weak association between sleep duration and anemia was still too conclusive to be drawn. Given large sample size, this weak association is not appealing and convincing.
Author Response
"I would like to extend my deep appreciation to you for review. I thankfully found this round of editing a good chance of improving the paper."

This manuscript is a resubmission of an earlier submission. The following is a list of the peer review reports and author responses from that submission.
Round 1
Reviewer 1 Report
Please find the attached file.

Author Response
I would like to extend my deep appreciation to you for review. I thankfully found this round of editing a good chance of improving the paper

Reviewer 2 Report
The authors examined the association between sleep duration and the risk of anemia using KNHANES data. Although the big sample size is a merit, the cross-sectional study design and the novelty and the implication/significance of the study is not well stated in the current version. There are issues as shown in the list:
- The conclusion of inverse association between sleep duration and anemia seems a bit forced to draw in pre-menopausal women because the estimates is quite small and CI is very close to 1. From model 1-5, even model 4 and 5 seem to have significant result, the CI is very close to 1, so it might just came out of a sudden or randomly. In fact, most of estimates are NOT significant in adjusting for multiple variates. In such a large sample, the CI is very close to 1, I would assume no association there. The authors seem to be very keen to present significant results, so even presented the only significant results from unadjusted models.
- Why did the authors have 5 sleep duration group? why can't 7 and 8 hours/day merged into one group as most studies did? What is the rationale/reference to do so?
- Why pregnant women included in the analysis?
- other minor things like it should be "multivariable" instead of "multivariate" analysis, as there is more than one covariates in the model, and only one outcome in your model.
Author Response

(The authors gave the same response as above.)

Round 2
Reviewer 1 Report
Please find an attached review.
